# Evaluation of micro rain radar-based precipitation classification algorithms to discriminate between stratiform and convective precipitation

Andreas Foth[1], Janek Zimmer[2], Felix Lauermann[1,3], and Heike Kalesse[1]

[1]Leipzig Institute for Meteorology, University of Leipzig, Leipzig, Germany
[2]Meteologix AG, Sattel, Switzerland
[3]now at: Deutscher Wetterdienst, Meteorologisches Observatorium Lindenberg/Richard–Aßmann–Observatorium, Tauche, Germany

**Correspondence:** Andreas Foth (andreas.foth@uni-leipzig.de)

**Abstract.** In this paper, we present two micro rain radar-based approaches to discriminate between stratiform and convective precipitation. One is based on probability density functions (PDFs) in combination with a confidence function and the other one is an artificial neural network (ANN) classification. Both methods use the maximum radar reflectivity per profile, the maximum of the observed mean Doppler velocity per profile and the maximum of the temporal standard deviation ($\pm 15\,\mathrm{min}$) of the observed mean Doppler velocity per profile from a micro rain radar (MRR). Training and testing of the algorithms were performed using a two year data set from the Jülich Observatory for Cloud Evolution (JOYCE). Both methods agree well giving similar results. However, the results of the ANN are more decisive since it is also able to distinguish into an inconclusive class, in turn making the stratiform and convective classes more reliable.

## 1 Introduction

Evaporation of precipitation below cloud base is a crucial process in the water- and energy cycle. Precipitation can be of two clearly distinguishable types – stratiform and convective. Both types originate from different clouds (Houze Jr, 2014). Stratiform precipitation mainly falls from nimbostratus whereas convective precipitation originates from active cumulus and cumulonimbus clouds. These cloud types may occur separately or entangled in the same cloud complex.

The parameterization of the precipitation evaporation process is highly empirical in current general circulation models (Rotstayn, 1997). Evaporation of precipitation generates cold pools that lead to convective organisation (Schlemmer and Hohenegger, 2014) and Tropical storms (Pattnaik and Krishnamurti, 2007); it is highly relevant for boundary-layer humidity (Worden et al., 2007) and subsequently for the Tropical general circulation (Bacmeister et al., 2006). However, also in the midlatitudes, precipitation evaporation is an important factor in the water cycle (Morrison et al., 2012) and the simulated water cycle pro-

cesses are highly sensitive to the empirical parameters and assumptions.

In order to improve the parameterization of evaporation from convective rain a big data set of convective rain cases is needed to generate robust statistics. Since it is a large effort to manually discriminate between stratiform and convective cases, automated algorithms were developed.

In previous approaches, stratiform and convective rain are separated based on the rain drop size distribution measured by a disdrometer (Caracciolo et al., 2006; Thompson et al., 2015; Ghada et al., 2019). Precipitation was also classified using radar images and radar wind profiler data (Rosenfeld et al., 1995; Williams et al., 1995; Tokay and Short, 1996; Tokay et al., 1999; Yang et al., 2013). Deng et al. (2014) classified convective precipitation based on thresholds of the radar reflectivity and the gradient of accumulative radar reflectivity retrieved from a vertically pointing cloud radar. Geerts and Dawei (2004) used a decision tree to separate different precipitation types by means of cloud radar variables. Additionally, discrimination algorithms using an ANN were developed (Yang et al., 2019; Ghada et al., 2019) . The ANN approach of Yang et al. (2019) is based on ground-based Doppler Radar observations. Lazri and Ameur (2018) combined a support vector machine, ANN and random forest to improve the stratiform convective classification using spectral features of SEVIRI data. Jergensen et al. (2020) classify thunderstorms into three categories: supercell, part of a quasi-linear convective system, or disorganized using radar data in a machine learning approach.

In summary, several approaches such as ANN, fuzzy logics, or decision trees based on different instruments such as disdrometer, cloud radar, precipitation radar, or radar wind profiler were developed in the past. In this paper, two methods are developed which classify rain as stratiform or convective event based only on MRR observations to enable a wide spread and straightforward usage for ground-based remote-sensing sites.

## 2 Instrumentation

### 2.1 Supersite JOYCE

In recent years, the Jülich Observatory for Cloud Evolution (JOYCE[1]) was equipped with a combination of synergistic ground-based instruments (Löhnert et al., 2015). JOYCE is situated at $50°54'31''$N and $6°24'49''$E with an altitude of 111 m MSL. In 2017 JOYCE was transformed into a Core Facility (JOYCE – CF) funded by the DFG (Deutsche Forschungsgemeinschaft) with the aim of high quality radar and passive microwave observations of the atmosphere. The supersite operates a variety of ground-based active and passive remote sensing instruments for cloud and precipitation observations, for example: X, Ka, and W-Band radars, ceilometers, a Doppler wind lidar, an atmospheric emitted radiance interferometer (AERI), a Sun photometer, disdrometers, several radiation measurement systems, as well as an MRR. The latter is the main instrument in this study and

---

[1]JOYCE webpage: http://cpex-lab.de/cpex-lab/EN/Home/JOYCE-CF/JOYCE-CF_node.html, last accessed: 2020-12-01

is explained in detail in the following sub-section. The data used in this study was gathered in 2013 and 2014. The data from 2013 covers the entire year and was used to train the algorithms (training data set). The data from 2014 covers almost the entire year apart from February. It is a completely independent data set and is used as test data set for the algorithms. In 2013 and 2014, 471 and 683 hours of rain were observed, respectively.

## 2.2 Micro rain radar

The micro rain radar (MRR) which is built by the Metek (Meteorologische Messtechnik GmbH) company, is a compact FM-CW (frequency modulated-continuous wave) Doppler radar operating at 24 GHz (Peters et al., 2002). The MRR at JOYCE (in 2013 and 2014) is an MRR-2 system operating with 32 range gates. The lowermost range gates (number 0, 1 and 2) up to 200 m are affected by near-field effects and the last range gate of 3100 m is too noisy. These range gates are usually omitted according to Maahn and Kollias (2012). Hence, 28 range gates from 300 to 3000 m remain for the analyses in this study. The vertical and temporal resolution amounted to 100 m and 1 min, respectively. The MRR data was processed according to Peters et al. (2005). The instrument was zenith pointing and measured the radar Doppler spectrum from which the mean Doppler velocity ($v_\mathrm{D}$) were derived. The radar reflectivity factor ($Z$) is derived via integrating over the drop size distribution according to Peters et al. (2005).

## 3 Stratiform convective discrimination

### 3.1 Convection indices

Several weather indices can be used to describe the stability of the atmosphere (Kunz, 2007). Three indices that are based on thermodynamic profiles are described in the following. All give a hint on the probability of convection based on COSMO (Consortium for Small-scale Modeling) EU model data. COSMO-EU has a horizontal resolution of 7 km and a vertical resolution between around 60 m and 370 m below 3 km. The temporal resolution amounts to 1 h. The weather index total totals is a combination of the vertical totals ($VT$) and cross totals ($CT$). The $VT$ is the temperature ($\vartheta$ in ° Celsius) difference between 850 hPa and 500 hPa while the $CT$ is 850 hPa dewpoint ($\tau$) minus the 500 hPa temperature:

$$
\begin{aligned}
TT &= VT + CT \\
&= (\vartheta_{850} - \vartheta_{500}) + (\tau_{850} - \vartheta_{500}).
\end{aligned}
\tag{1}
$$

The higher the $TT$, the more probable is convection.

The second index, named KO index (Andersson et al., 1989), describes the potential instability between lower and higher levels of the atmosphere (at 1000 hPa, 850 hPa, 700 hPa, and 500 hPa). It is thus based on the pseudo-potential temperatures $\theta_e$:

$$
KO = 0.5\,(\theta_{e,700} + \theta_{e,500} - \theta_{e,1000} - \theta_{e,850}).
\tag{2}
$$

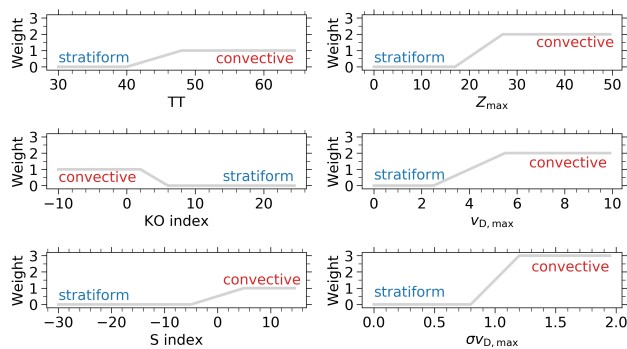

**Figure 1.** Weight of the meteorological and radar-based convection criteria: total totals ($TT$), convection index ($KO$), soaring index ($S$), maximum of the reflectivity ($Z_{max}$) per profile, maximum of the Doppler velocity ($v_{D,max}$) per profile and the maximum (per profile) of the temporal standard deviation of the Doppler velocity ($\sigma v_{D,max}$) within a $\pm15\,\mathrm{min}$ interval.

The lower the $KO$ index the higher the potential of convection.

The soaring index[2] ($S$) is intended to be a tool in soaring and sporting aviation because it gives a hint on thermal lift and hence on instability. It is defined as:

$$S \quad = \quad \vartheta_{850} - \vartheta_{500} + \tau_{500} - (\vartheta_{700} - \tau_{700}). \tag{3}$$

The higher the $S$ index the higher the probability of convection.

### 3.2 Convection score

First, a convection score to classify three types of precipitation labelled as stratiform, convective and inconclusive, is defined by applying a threshold range to six different variables. Three variables are based on thermodynamic profiles ($TT$, $KO$, $S$) and three are based on the MRR observations. Specifically, the used MRR variables are: the maximum of reflectivity ($Z_{max}$) per profile, maximum of the mean Doppler velocity ($v_{D,max}$) per profile and the maximum (per profile) of the temporal standard deviation ($\pm15\,\mathrm{min}$) of the mean Doppler velocity ($\sigma v_{D,max}$). The profile maxima are calculated between ground and $3\,\mathrm{km}$. It is expected that larger rain drops are usually caused by convective precipitation (Niu et al., 2010) which leads to higher $Z$ and $v_D$ values, respectively. Furthermore, stratiform precipitation is expected to be less variable over time whereas convective precipitation results in a larger standard deviation of $v_D$ over time. It is assumed that $\pm15\,\mathrm{min}$ is a reasonable time span for classification of rain events. If the rain event is shorter than $30\,\mathrm{min}$ ($\pm15\,\mathrm{min}$) the variability is determined over this shorter period. The maxima of the height dependent $Z$, $v_D$ and $\sigma v_D$ are used to assign the vertical properties to profile properties. In case of cold stratiform rain there might be a clearly defined melting layer. The so-called radar 'bright band' is indicated by erroneously high reflectivity values $Z$ in the layer of melting ice particles which force the detection to be convective. Therefore, two other variables ($v_D$ and $\sigma v_D$) are chosen which are not affected by the melting layer and both will counteract the false

---

[2]http://www2.wetter3.de/soaring_index.html, last accessed: 2020-12-01

classification and force the retrieval to classify stratiform.

Different weightings are assigned to the six variables as visualized in Fig. 1. Whenever a variable exceeds a convection threshold range (or falls below in case of KO index), the weight to be convective increases. A detailed description of the value range of convection indices such as $TT$ and $KO$ is given by (Kunz, 2007). The threshold range of the MRR-based variables is determined empirically. For this purpose, the variables were closely examined and adjusted for clearly evident cases. The weightings of all variables are summed up resulting in the convection score which ranges from $0$ to $10$. The application of a smooth linear threshold range of weights between stratiform and convective is more realistic and leads to a more homogeneous distribution of the convection score compared to using strict binary thresholds. By using six variables the classification is more robust against false classifications than those based on one single variable. The model-based variables have lower weightings because they are based on temperature and humidity profiles from the nearest COSMO-EU model grid cell which may differ crucially in whether it is a precipitating or a non-precipitating grid cell. Therefore, the model derived measures ($TT$, $KO$, and $S$) have lower weights, because the measured MRR variables are more trustworthy than the modeled ones. $\sigma v_{\mathrm{D,max}}$ is assigned to have the highest weight because the variability of the rain intensity is assumed to be the best criterion for the stratiform-convective discrimination.

Figure 2 illustrates the distribution of the convection score. Whenever a convection score is less than 3 the profile is assigned to be stratiform. Values between 3 and $5.5$ are stated as inconclusive. Values larger than or equal to $5.5$ are assigned as convective. These strict thresholds enable a very certain classification with a low amount of false classifications. The inconclusive zone between stratiform and convective indicates a transition between both. The thresholds of 3 and $5.5$ were chosen to confidently separate two classes which are mainly free of false classified rain events resulting in a confident data set for training the algorithms. This approach replaces a manual inspection by visual classification of each single profile. However, several rain events (approx. $10\,\%$) were reviewed by eye to verify a correct classification. That means randomly selected cases were checked if the convection score worked as intended and the synoptic situation was reviewed.

At this step, each profile is either classified as stratiform, inconclusive, or convective using the convection score and this assignment is stated as true state to train the algorithms explained below. Since the motivation of this work is to classify the precipitation type and its confidence purely based on the MRR observations the following methods based on PDFs or ANN are developed. Since the PDF and ANN method are based on training, the MRR data has to be free of extreme or unphysical values. Therefore the MRR data (input) is filtered. Only measurements with $Z_{\mathrm{max}}$ between -10 and $50\,\mathrm{dBZ}$, $v_{\mathrm{D,max}}$ between 0 and $10\,\mathrm{m\,s}^{-1}$ and $\sigma v_{\mathrm{D,max}}$ between 0 and $2.5\,\mathrm{m\,s}^{-1}$ are taken into account.

Here, the question might arise why inconclusive profiles should be learned by algorithms. In fact, rain events can be ambiguous and cannot be classified into stratiform or convective, especially stratiform rain moving towards mountainous area which causes convection. On the other hand, vertical air motion and turbulence influence $v_{\mathrm{D,max}}$ and might shift stratiform

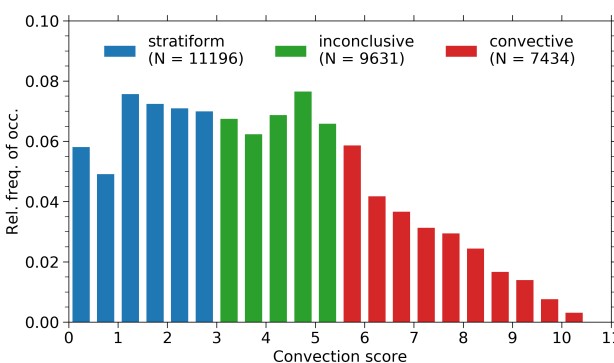

**Figure 2.** Relative frequency of occurrence of the convection score.

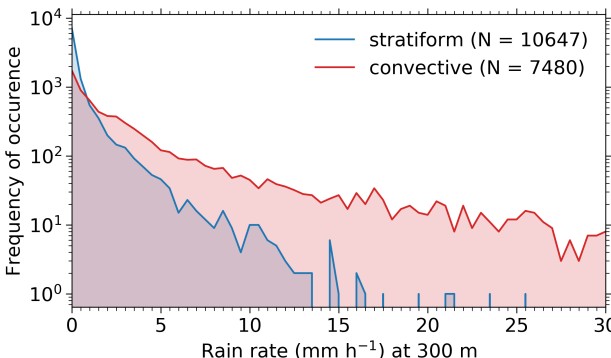

**Figure 3.** Frequency of occurrence of rain rate at 300 m height for stratiform (blue) and convective (red) rain cases.

profiles towards higher convection scores and convective profiles to lower scores. A class with inconclusive profiles accounts for the mentioned features and avoids misclassifications into the stratiform and convective classes, respectively.

The frequency distribution of rain rate at 300 m height is shown in Fig. 3. The precipitation cases are separated by the convec-
5   tion score. The stratiform precipitation mostly causes low rain rates below $1\,\mathrm{mm\,h^{-1}}$ whereas high rain rates above $15\,\mathrm{mm\,h^{-1}}$ are very rare. In contrast, high rain rates above $15\,\mathrm{mm\,h^{-1}}$ are caused by convective precipitation. It has to be considered that the absolute number of occurence differ from Fig. 2 because precipitation disappears due to evaporation on the way through the atmosphere and is not reaching 300 m which is the lowest available MRR height.

10  **3.3    Rain classification method based on PDF**

This algorithm was developed based on the classification algorithm by Liu et al. (2004) which was originally developed for the Cloud-Aerosol Lidar and Infrared Pathfinder Satellite Observations (CALIPSO) aerosol cloud discrimination (Winker et al.,

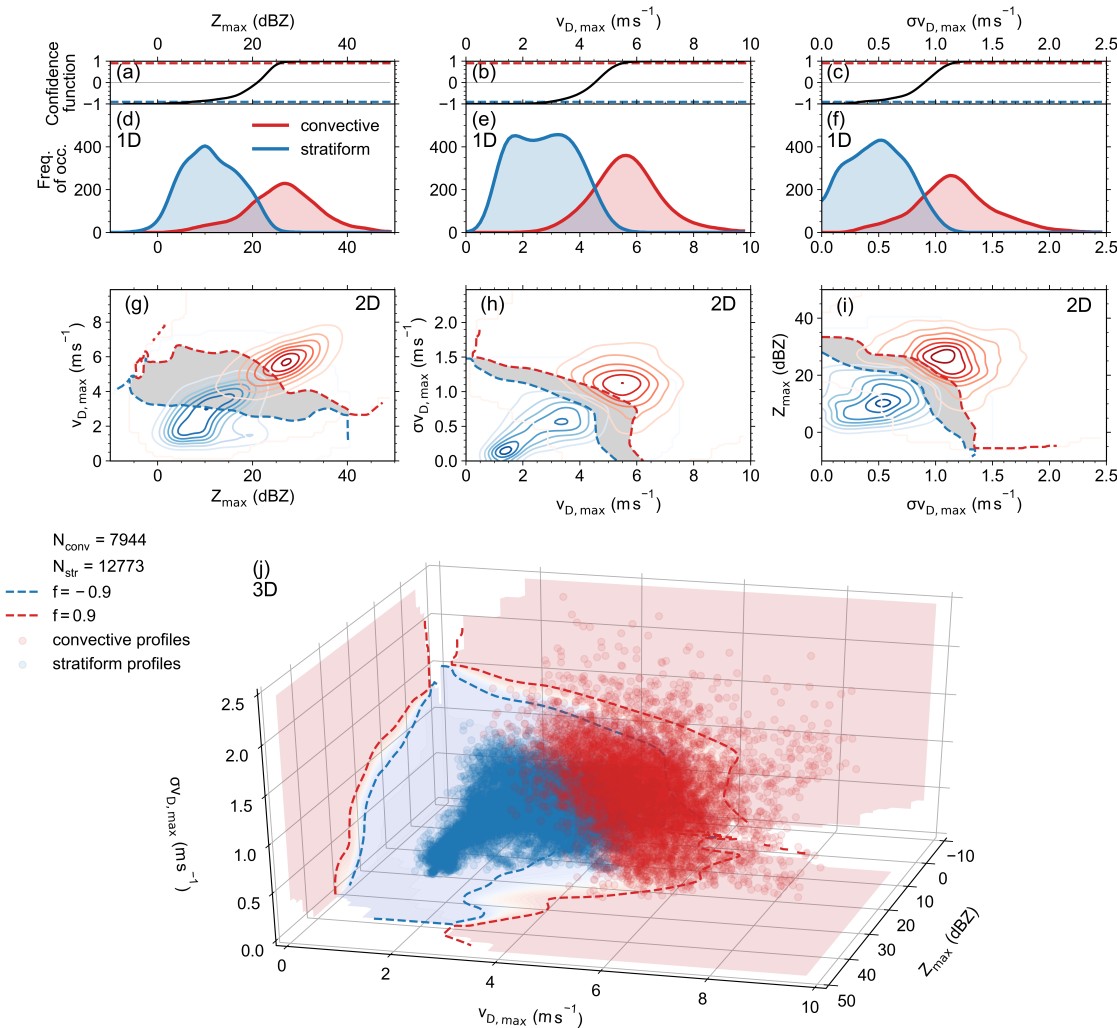

**Figure 4.** Overview about the one-dimensional (1D:d,e,f), two-dimensional (2D:g,h,i), and three-dimensional (3D:j) probability density functions for the maximum radar reflectivity $Z_{max}$ per profile (d), the maximum of the observed Doppler velocity $v_{D,max}$ per profile (e), the maximum of the temporal standard deviation of the observed Doppler velocity $\sigma v_{D,max}$ per profile (f), and each 2D combination of these three variables (g-i). (j) shows the 3D scatterplot of the three variables with the contour of the corresponding confidence values in each plane. (a-c) show the confidence function of the corresponding 1D distributions. The dashed lines represent the thresholds of a confident classification with values beyond -0.9 and 0.9, whereas values in between are indicated by a grey area (g-i). Stratiform or convective profiles are indicated by blue or red colours and by low or high values of the confidence functions, respectively.

2009). It shows that the confidence of a discrimination algorithm can be improved by using three measurement variables instead of only one or two. Later on, Liu et al. (2009) improved the algorithm by using five instead of three variables. Here, this separation approach is modified for MRR variables to classify precipitation into stratiform or convective.

The confidence function is defined as:

$$f(X) = \frac{n_{\mathrm{c}}(\boldsymbol{X}) - n_{\mathrm{s}}(\boldsymbol{X})}{n_{\mathrm{c}}(\boldsymbol{X}) + n_{\mathrm{s}}(\boldsymbol{X})} \tag{4}$$

$$= \frac{P_{\mathrm{c}}(\boldsymbol{X}) - P_{\mathrm{s}}(\boldsymbol{X})/N_{\mathrm{s}}/N_{\mathrm{c}}}{P_{\mathrm{c}}(\boldsymbol{X}) + P_{\mathrm{s}}(\boldsymbol{X})/N_{\mathrm{s}}/N_{\mathrm{c}}} \tag{5}$$

5   with $n_i$ being the number of occurrences of class $i$ (stratiform s or convective c) having attribute $X$ and $N_i$ the total number of events for the $i$th class. $P$ is the PDF of $\boldsymbol{X}$ which can be multidimensional $\boldsymbol{X} = [X_1, \ldots, X_m]$. The used bin size of the distributions of $Z$, $v_{\mathrm{D,max}}$, and $\sigma v_{\mathrm{D,max}}$ amounts to $0.5\,\mathrm{dB}$, $0.125\,\mathrm{m\,s}^{-1}$, and $0.025\,\mathrm{m\,s}^{-1}$, respectively. The value of $f$ is bounded by [-1,1]. The lower the value, the more probable the MRR-observed rain profile is of stratiform nature. Values of exactly $-1$ are treated as certainly stratiform and values of $+1$ correspond to certainly convective profiles. Values around 0 indicate uncertain

10   classifications. On the basis of the return value of $f$ a classification and a measure of the confidence of this classification can be derived. The sign of $f$ determines the class assignment and the absolute magnitude of $f$ assigns the confidence to the classification. In the following the PDFs are smoothed using a Gaussian filter with a standard deviation of 3 bins in each dimension to account for gaps in the PDF due to missing variable combinations in the training data. Applying such a Gaussian filter makes the PDF method more robust.

  Figure 4 (d), (e) and (f) show the one-dimensional distribution of the three MRR variables ($Z_{\mathrm{max}}$, $v_{\mathrm{D,max}}$ and $\sigma v_{\mathrm{D,max}}$) and their corresponding confidence functions $f$ (a,b,c). Here, the stratiform and convective precipitation profiles are distinguished by the convection score explained above. However, $Z_{\mathrm{max}}$ at values between 5 and $25\,\mathrm{dB}$ shows a region of overlap between both classes resulting in low magnitude of $f$ ranging between $-0.8$ and $0.8$. $Z_{\mathrm{max}}$ below $5\,\mathrm{dBZ}$ or above $25\,\mathrm{dBZ}$ can be reliably

20   classified as stratiform or convective, respectively (Fig. 4 d). The distribution of $v_{\mathrm{D,max}}$ (Fig. 4 e) as well as $\sigma v_{\mathrm{D,max}}$ (Fig. 4 f) show significant overlap regions between stratiform and convective profiles between 2.5 and $5.5\,\mathrm{m\,s}^{-1}$ and between 0.4 and $1.1\,\mathrm{m\,s}^{-1}$, respectively. This results in absolute magnitudes of $f$ below 0.8 for both $v_{\mathrm{D}}$-related variables. The overlap area of stratiform and convective profiles is smallest for $\sigma v_{\mathrm{D,max}}$ but since vertical air motion and turbulence influence $v_{\mathrm{D,max}}$, it can not serve as stand-alone value. In conclusion, a classification algorithm based on only one of the mentioned MRR variables is not

25   able to unambiguously distinguish between stratiform and convective precipitation indicated due to existing overlap regions.

  The ambiguity can be reduced by adding a second dimension to the PDF. Figure 4 (g), (h) and (i) illustrate the distribution of each two-dimensional (2D) combination of the three MRR-based variables. The dashed lines indicate the $f$ values of $-0.9$ and $0.9$. The values in between represent the overlap where no unambiguous assignment can be made (grey area). The peaks of the

30   two classes are clearly separated for all three variable combinations (g,h,i). Nevertheless, there are still observations leading to an ambiguous assignment. In principle, these ambiguous assigned profiles with $f$ values between $-0.9$ and $0.9$ could be stated as inconclusive. However, the PDF algorithm is not trained to classify inconclusive cases. A quantitative estimation of how well the discrimination works is given at the end of this sub-section.

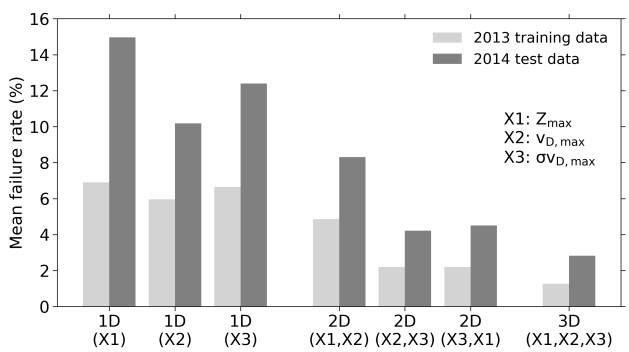

**Figure 5.** Comparison of the mean stratiform convective discrimination failure rates for two data sets, the training data set from 2013 (light grey) and the test data set from 2014 (dark grey). The shown failure rate is separated for PDFs with increasing dimension: 1D is based on only one MRR variable, 2D is a two dimensional PDF based on two MRR variables and, 3D is a three dimensional PDF based on three MRR variables, as mentioned in the legend.

By using all three mentioned MRR-based variables a three-dimensional (3D) PDF can be created which is visualized in Fig. 4 (j). It is indicated that both stratiform and convective profiles are clearly separated with a very small region of overlap. The quality of the 3D PDF-based classification in contrast to 2D and 1D can be explained in terms of failure rates $R_{\mathrm{f}}$ (Liu et al., 2009):

$$R_{\mathrm{f}}(\boldsymbol{X}) \quad = \quad \frac{|f(\boldsymbol{X}) - 1|}{-2}. \tag{6}$$

As explained above the performance of the classification is limited by the amount of overlap in the PDFs. The smaller the overlap, the more clear is the separation between stratiform and convective profiles. Figure 5 presents the mean failure rate for the 1D-, 2D-, and 3D PDFs of the training data set 2013 and the independent test data set 2014. The training data was used to build the PDF for the calculation of $f$. For each profile from the test data, the according $f$ value can be read out from the trained confidence function. To account for measurement uncertainties or turbulence influencing all radar variables, the $f$ underlying PDFs are smoothed using a Gaussian filter with a standard deviation of 3 bins in each dimension corresponding to a $Z_{\mathrm{max}}$ of 1.5 dB, $v_{\mathrm{D,max}}$ of $0.38\,\mathrm{m\,s^{-1}}$, and $\sigma v_{\mathrm{D,max}}$ of $0.08\,\mathrm{m\,s^{-1}}$. In Fig. 5 it is obvious that a reduction of the overlap by adding another attribute (MRR-based variable) results in smaller failure rates. Highest failure rates result from the 1D-PDFs. The mean failure rate for the 3D-PDF based rain classification discrimination for training and test data is less than 1 % and 3 %, respectively. This is much lower than the failure rates of 1D and 2D PDFs for stratiform-convective discrimination, which range between 2 to 7 % for the training data set and 3 to 15 % for the test data set.

It was shown that the algorithm performance could be improved by adding more variables. However, the amount of independent variables only obtained by MRR is limited. $Z$ calculation is based on the drop number concentration. Other MRR variables such as rain rate or liquid water content are also based on drop number concentration and are hence not independent

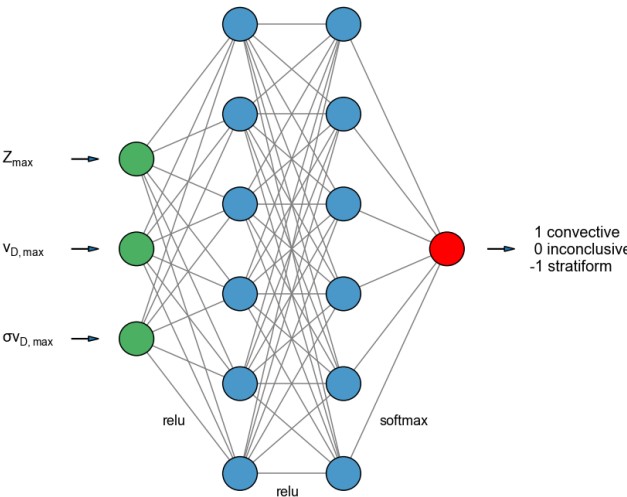

**Figure 6.** Diagram of the neural network with an input layer consisting of three nodes (green) according to the three MRR-based variables, two hidden layers with six nodes each (blue) and the output layer with one node (red).

from $Z$ and would not add any more information to the discrimination algorithm.

## 3.4 Method based on an artificial neural network (ANN)

A classification of rain as stratiform, convective or inconclusive can also be based on an ANN. The ANN model created is a multi-layer perceptron approach implemented using the open source machine learning library for research and production *TensorFlow*[3]. It is trained with $Z_{\max}$, $v_{D,\max}$ and $\sigma v_{D,\max}$ from the training data set (2013). The ANN model (Fig. 6) consists of 3 input nodes ($Z_{\max}$, $v_{D,\max}$ and $\sigma v_{D,\max}$) and is further composed of two hidden layers with 6 nodes each, and one output node to learn how to classify rain events according to the true classification made by the convection score. With the chosen number of hidden layers and nodes, nonlinear relationships can be better represented and the ANN shows best performance. Giving $Z_{\max}$, $v_{D,\max}$ and $\sigma v_{D,\max}$ as input to the ANN it will classify the rain event with probabilities to be stratiform (labeled as $-1$), inconclusive (0), and convective (1). To finally classify the rain event the class with the highest probability is stated to be the actual rain type. For example, the ANN outputs the probabilities 0.7 for stratiform, 0.2 for inconclusive, and 0.1 for convective. Then this profile is classified to be stratiform with the label $-1$. The model is trained for 500 epochs (iteration steps) and the training data is shuffled before each epoch. The algorithm *Adam*[4] is used to optimize the model. As activation functions relu (rectified linear unit) and softmax are used for the two hidden layers and the output layer, respectively. Relu avoids negative output whereas softmax produces an output which is a range of values between 0 and 1, with the sum of the probabilities been equal to 1. As loss function the categorical cross-entropy is used to compute the cross-entropy loss between the truth and the

---

[3]TensorFlow: An end-to-end open source machine learning platform, https://www.tensorflow.org/, last accessed: 2020-12-01

[4]Kingma, D.P. and Ba, J.: Adam: A Method for Stochastic Optimization, 2014. web: https://arxiv.org/abs/1412.6980, last accessed: 2020-12-01

predictions. The cross-entropy is a measure of the difference between two probability distributions. The accuracy of the neural network can be described in terms of how often the predictions equal the truth. The ANN accuracy of the independent test data set (2014) amounts to 80 %.

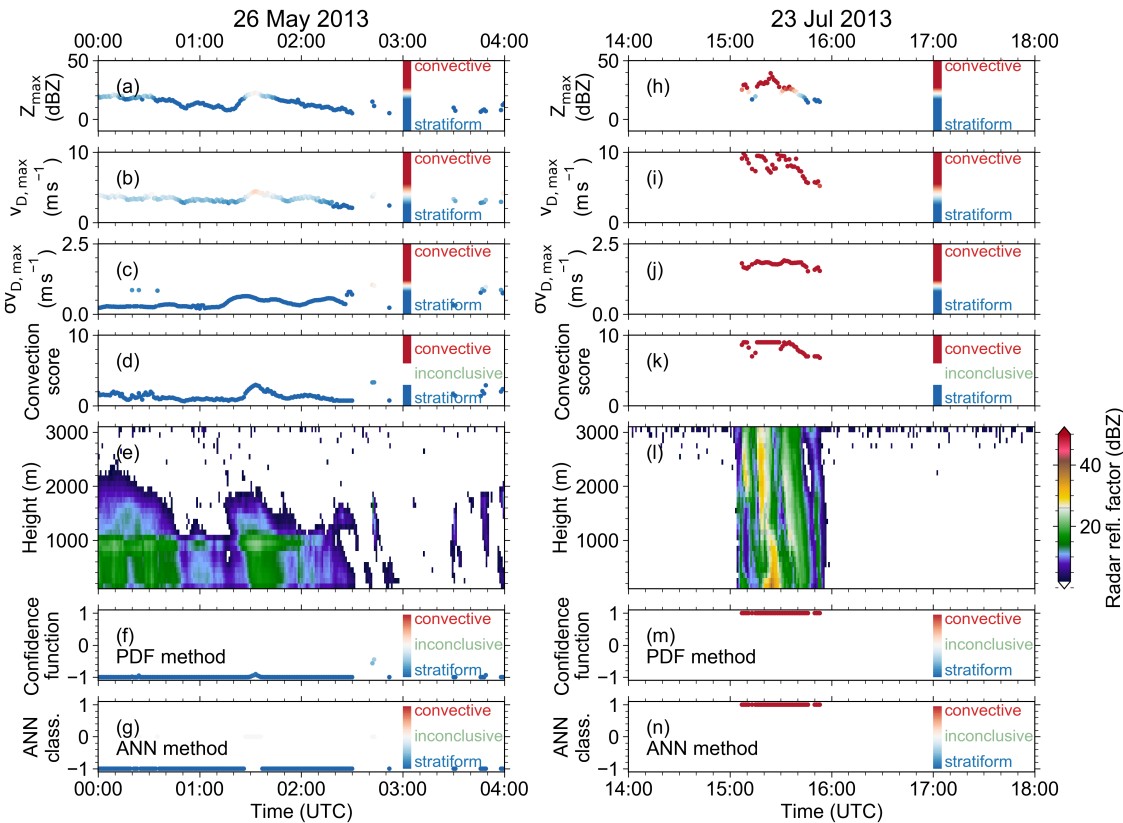

**Figure 7.** Discrimination indices: $Z_{max}$ (a,h), $v_{D,max}$ (b,i), $\sigma v_{D,max}$ (c,j) and the convection score indicating the true rain type (d,k). MRR reflectivity (e,l) and the rain classification based on the PDF method given by the confidence function (f,m) and based on the ANN (g,n). The data points in the time series are coloured according to the convection criteria whether the values indicate stratiform or convective rain. Left panels refer to the case study of 26 May 2013, right panels show 23 July 2013.

## 4 Results

After the successful development and evaluation of the classification algorithms, both the 3D-PDF-based and the ANN were applied to two case studies. The first one was a rainy night on 26 May 2013 (Fig. 7 a-g). Figure 7 (e) shows the time–height display of the radar reflectivity factor. The day began with rain from 00:00 UTC to 02:30 UTC. The rain fell homogeneously

5    with only small variations in $Z_{max}$ (a), $v_{D,max}$ (b) and $\sigma v_{D,max}$ (c). The calculated convection score (d) was very low which means that these rain events were stated to be stratiform. For this springtime rain events, the PDF (f) and ANN (g) method produce very similar results and both agree with the true class given by the convection score.

The right panel of Fig. 7 shows the same quantities as on the left panel but for 23 July 2013. This case indicates convec-

10    tive rain falling between 15:00 UTC and 16:00 UTC. $Z_{max}$ (h), $v_{D,max}$ (i) and $\sigma v_{D,max}$ (j) and the calculated convection score

are characterized by high values representing convective rain. Figure 7 (l) shows the radar reflectivity factor of the shower. The PDF- and ANN method are in a very good agreement and classify each profile as convective in conformity with the convection score (truth).

The performance of both algorithms over a whole year (test data year 2014) is illustrated in Fig. 8. It shows the relative frequency of occurrence of precipitation profiles that are defined by the convection score (truth) to be stratiform (a), inconclusive (b), or convective (c). For the PDF method cases are stated as stratiform when the $f$ value is lower than $-0.9$, inconclusive when $f$ is between $-0.9$ and $0.9$, and convective when $f$ is larger than $0.9$. For the stratiform cases the PDF and ANN methods classify most stratiform cases to be stratiform ($84.7\%$ and $96.1\%$ for ANN and PDF, respectively, see Fig. 8 a). Only $15.3\%$ and $3.9\%$, respectively, are erroneously classified as inconclusive. These are cases with higher convection scores with averaged values around roughly 2.5 which is closer to the transition of convection scores larger than 3 that are stated as inconclusive. As expected, neither ANN nor PDF misclassified true stratiform cases as convective. The performance of the classification of true convective cases (c) is very similar. There are almost no completely misclassified cases and only a few percent of erroneously inconclusive cases. Here the averaged convection scores are roughly 6 which means on the lower edge of the convective classification and close to the transition of convection scores of less than 5.5 that are stated as inconclusive. $85.8\%$ and $98.1\%$ (ANN and PDF) of the true convective cases are correctly classified as convective.

However, the most critical point is the classification of the true inconclusive cases (Fig. 8 b). Only $71\%$ and $35.8\%$ (ANN and PDF) are correctly classified. That means that nearly $30\%$ of ANN-based profiles and nearly $65\%$ of PDF-based profiles of all true inconclusive cases are classified as stratiform or convective. The ANN is performing better here. This is caused by the strict convection score discrimination which was stated as truth. In fact, these inconclusive cases might be classified as stratiform or convective but the thresholds were chosen very strict to confidently separate two classes which are mainly free of misclassified rain events. The averaged convection score of the false stratiform inconclusive cases (ANN) amounts to 3.6 and of the false convective inconclusive amounts to 5 (Fig. 8 b). The erroneously stratiform and convective classified inconclusive cases of the PDF method ($25.9\%$ and $38.3\%$) has averaged convection score values of 3.6 (stratiform) and 4.9 (convective). Apparently, these cases would be correctly classified in case of less strict convection score thresholds than currently used (3 and 5.5, see Fig. 2). It is expected to improve the ANN and PDF performances by gathering more data for algorithm training.

It has to be considered that the total amount of data is different for both methods. This is due to the fact that some combinations of the three input variables do not appear within the training data causing gaps in the 3D-PDF. Those combinations cannot be classified but its amount is less than $0.2\%$ for the 2014 test data set.

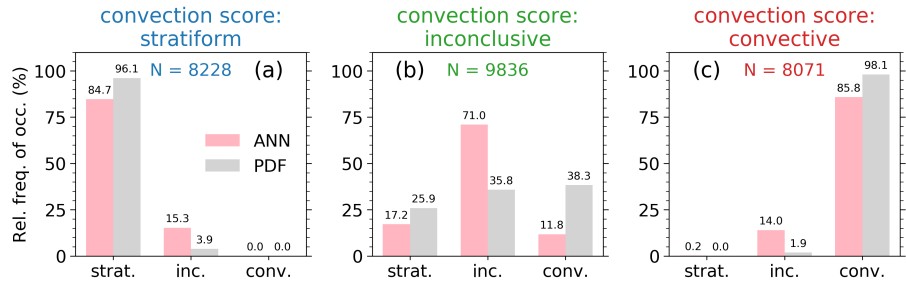

**Figure 8.** Relative frequency of occurrences of stratiform, inconclusive and convective rain classification for both ANN (light pink) and PDF (grey) method separated by the true class (a,b,c) based on the test data 2014. The true class is defined by the strict convection score discrimination. The colored numbers denote the sample size. The numbers on top of each bar indicate the actual value.

## 5 Conclusions and outlook

In order to improve microphysical parametrizations within small-scale models one has to deal with large data sets. The presented rain type classification methods based on PDF and ANN algorithms are suited to process micro rain radar data from long time series. The effort of creating a robust training data set without unphysical data between both methods is similar and the application of both methods is straightforward. The main advantage of the ANN in contrast to the PDF method is that the ANN method was trained to directly classify inconclusive profiles which leads to a lower amount of false classified profiles.

In a next step, evaporative cooling rates will be estimated for convective rain events to parametrize the cooling by means of temperature, relative humidity and rain droplet number concentration. It is also planned to apply the algorithms to different ground-based remote-sensing sites that have long-term MRR observations to create stratiform-vs-convective rain event climatologies. At present, the new MRR of the University of Leipzig[5] is running 24/7. In the near future, the classification algorithms will be applied operationally and will be improved with continuously gathered data.

*Code availability.* The open source machine learning library for research and production TensorFlow (Abadi et al., 2015) used for this publication is available under https://www.tensorflow.org/, last accessed: 2020-12-01.

*Author contributions.* AF prepared the manuscript in close collaboration with HK. AF performed the investigations and data analyses. JZ and FL contributed with their knowledge about basic meteorology and convective indices. JZ also contributed with his experience in radar data analysis. The conceptualization was initialized by AF. All authors have contributed to the scientific discussions.

---

[5]https://home.uni-leipzig.de/remsensarctic/index.php?lang=en&content=ql, last accessed: 2020-12-01

*Competing interests.* The authors declare that they have no conflict of interest.

*Acknowledgements.* This research has been supported by German Science Foundation (DFG) through funding the grants FO 1285/2-1, KA 4162/2-1 (PICNICC within the priority programme PROM) and WE 1900/33-1 and the Federal Ministry of Education and Research in Germany (BMBF) through High-Definition Clouds and Precipitation for advancing Climate Prediction research programme (HD(CP)2; FKZ: 01LK1504C and 01LK1502N). Data from JOYCE were provided by DFG-funded Core Facility (JOYCE-CF) under DFG research grant LO 901/7-1.

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
