# Peer review of "Evaluation of micro rain radar-based precipitation classification algorithms to discriminate between stratiform and convective precipitation"

_Atmospheric Measurement Techniques, 2020_

## Referee Comment (RC1) · Anonymous Referee #1 · 20 Sep 2020

This study proposes two algorithms (PDF and ANN) for convective and stratiform precipitation separation based on the MRR measurements. The manuscript has a clear structure and smooth expression, but have some issues (e.g., weak literature survey, validation of results, the application value etc) and requires a major revision before its acceptance.

Detailed comments are provided below.

P1: In Introduction section, the authors should provide background on convective and stratiform rain in meteorological applications (e.g. Houze 2014). What are the existing methods for convective and stratiform rain separation? How the proposed algorithms

[Figure]

on convective and stratiform rain separation has the advantage over the existing different methods? Literature survey for the artificial neural network (ANN) for rain type classification is required. The novelty of the work is not enough highlighted.

Houze, R. A., Jr. (2014) Cloud Dynamics. Academic Press.

P3: Description of MRR is insufficient,. Please elaborate especially the signal processing part. The Ku-band signal attenuate/extinct in convective rain. How the authors make sure about this phenomena. What signal-to-noise ratio has been considered for processing the MRR dataset

P4: Ln 4: In stratiform case, the Zmax will be at the melting layer (bright band). Do the authors consider this factor in their analysis? Upto what height, the analysis is performed?

P4: Ln 8: On what basis 15 min time interval is taken?

P4: Ln11: On what basis the scores/weight are defined in Figure 1? Is threshold values of weight are region-specific? Which dataset is used to calculate the soaring index (S), convection index (Ko), total totals (TT)? What is the temporal and spatial resolution of those data?

P4: Ln 21: On what basis the convection score partition (stratiform less than 3, inconclusive 3-6 and convective >6) is taken? Does the author consider the rain rate criteria also?

P5: Figure 2: The inconclusive data points are more than the convective and stratiform samples. Please comment on it? Whether the inconclusive samples are the transition from convective to stratiform event.

P5: Ln 3: ….visual classification of each single profile. How the authors have visually classify the profile into convective and stratiform? What parameter and criteria are used for the visual classification? Please provide a skill score table for better representation of your results.

P10: Figure 6: Out of two proposed algorithms (PDF and ANN), which method is superior? Authors also need to discuss the source of errors for each method.

In the manuscript, the evaluation of the precipitation classification algorithms is not shown.

Please provide some discussion on the proposed algorithm and related future research to put the results into a broader context.

Minor:

P2: Ln25: …. following section. Change to sub-section.

P5: Ln 10: Zmax upto 50 dBZ. Don't you think there will be attenuation at such high reflectivity value?

P6: It will be good to show the rainfall distribution like figure 3d.

P7: What bin size the authors have considered for Eq. (4) and (5).

P11: Figure 7: PDF is overestimating the convective and stratiform precipitation than ANN. Which result is more accurate. For the inconclusive sample, both the methods have the same occurrence frequency. Why the number of data sample (NPDF and NANN) for analysis are different

---

## Referee Comment (RC2) · Anonymous Referee #2 · 22 Sep 2020

Title: "Evaluation of micro rain radar-based precipitation classification algorithms to discriminate between stratiform and convective precipitation" Authors: Andreas Foth and co-authors

General comments:

The study discusses the two algorithms PDF and ANN for classifying convective and stratiform precipitation profiles based on MRR data. The authors utilizes the maximum reflectivity, mean Doppler velocity and maximum deviation in velocity within +/- 15 min. But there have been a numerous studies on this topic using various ANN based algorithms (e.g., Ghada et al., 2019, doi:10.3390/atmos10050251; Jergensen

et al., 2020, DOI: 10.1175/WAF-D-19-0170.1). The paper is topic of interest. The authors presented only two algorithms. It could have been good to show the results from various ANN based models to discriminate the convective and startiform profiles and compare them. Further, authors should include the validation metrics such as RMSE, MAPE, etc in tabular form for both the models. Perhaps, use of convolution neural networks (CNNs), Long-short Term Memory (LSTM) and recurrence neural networks (RNNs) will provide better forecast for time series data. However, I concern about following comments. I recommend that this manuscript requires major revision before its acceptance.

Detailed comments are provided below:

P2: Why authors are used two year data for training? Is this data covers the all dynamic ranges observed convection/stratiform? Any ANN based model, the training data should have the all range of values.

P3: Are three indices such soaring index (S), convection index (Ko), total totals (TT) derived using COSMO model data? If so, is COMSO derived indices are validated with indices calculated form radiosonde observations?

P3:L5: Why the authors are used 15 minutes interval, where MRR gives 1 minute data?

P4.L6-7: . . .. . . convective precipitation contains larger rain drops . . . . .. Is it true always? Include reference.

P4.L8-9: . . .. . .. +/-15 min is a reasonable time span for classification of rain events. . .. . .But, there are the occasions, where the life time of convection will be less than 15 minutes. Authors should modify the sentence. Include reference.

P5.L9: . . ...PDF and ANN method are based on training, the data has to be free of extreme or unphysical values. . . .....Do authors mean that the data cleansing? I understood that the data filter was performed in MRR data. If so, rewrite the sentence. However, what are the extreme values? Because, in general, if the trained data consists of all

dynamics range, then the model will be able to predicted with better accuracy.

P5:L24: What are the modification are done in Liu et al. (2004) and Liu et al. (2009) algorithms.

P6:L23-24: ……. the confidence of a discrimination algorithm can be improved by using three measurement variables instead of only one or two….. Are the proposed number of variables are sufficient? Is the model predicting better accuracy, if you consider more than three input variables? Is the prediction depends on number of depended variables? Why authors are not consider rain rate for training MLP model?

P8:L16: Is the MLP model is multivariate multi-step? I also suggest including a table with hyperparameters of MLP used in this study, rather than mentioned in the text.

P9: For readers, change line colour to red in figure 6 (m) & (n).

P9: I would rather suggest to include about "how often convective/stratiform/inconclusive profiles occur at JOYCE supersite?

P11: The relative occurrences of inconclusive profiles are equally weighting with convective/stratiform in this study? Are they meant for transition profiles?

---

## Author Comment (AC1) · 1 Dec 2020

The comment was uploaded in the form of a supplement:
https://amt.copernicus.org/preprints/amt-2020-290/amt-2020-290-AC1-
supplement.pdf

---

## Author Comment (AC2) · 1 Dec 2020

**Response to Reviewers #1 and #2**

We like to thank the reviewers for providing helpful comments to improve the manuscript.

We made substantial changes according to your suggestions. All changes are highlighted in the diff-mansucript below. Added text is wavy-underlined and blue, discarded text is struck out and red. There are also minor changes in some figures that are not highlighted in the diff-mansucript below. Additionally, we slightly changed the algorithms and improved the performance. Therefore, some numbers changed in the manuscript. Furthermore, we replaced the stratiform case.

The reviewer comments are listed below in black. The authors response is written in blue.

**Anonymous Referee #1**

**General comments:**

This study proposes two algorithms (PDF and ANN) for convective and stratiform precipitation separation based on the MRR measurements. The manuscript has a clear structure and smooth expression, but have some issues (e.g., weak literature survey, validation of results, the application value etc) and requires a major revision before its acceptance.

**Detailed comments are provided below.**

P1: In Introduction section, the authors should provide background on convective and stratiform rain in meteorological applications (e.g. Houze 2014). What are the existing methods for convective and stratiform rain separation? How the proposed algorithms on convective and stratiform rain separation has the advantage over the existing different methods? Literature survey for the artificial neural network (ANN) for rain type classification is required. The novelty of the work is not enough highlighted.

Houze, R. A., Jr. (2014) Cloud Dynamics. Academic Press.

We expanded the literature review in the introduction.

P3: Description of MRR is insufficient. Please elaborate especially the signal processing part. The Ku-band signal attenuate/extinct in convective rain. How the authors make sure about this phenomena. What signal-to-noise ratio has been considered for processing the MRR dataset

SNR is not used. In case of strong convective rain with huge drops the measured reflectivity is high and may be attenuated by the drops. Nevertheless the reflectivity is expected to be too high to erroneously classify the rain to be stratiform. We added this explanation in sub-section 3.2.

P4: Ln 4: In stratiform case, the $Z_{max}$ will be at the melting layer (bright band). Do the authors consider this factor in their analysis?

Therefore we use three variables instead of only one. In case of cold stratiform rain with high $Z_{max}$ values in the melting layer the other two variables which are not affected by the bright band will force the retrievals (PDF and ANN) to classify stratiform rain. We added this explanation.

Up to what height, the analysis is performed?

The analysis is performed up to 3 km. We added this explanation.

P4: Ln 8: On what basis 15 min time interval is taken?

It is a reasonable time span for convective rain. If the rain shower is shorter than 30 min (plus minus 15 min) the shorter time span is used to derive the variability. We added this explanation.

P4: Ln11: On what basis the scores/weight are defined in Figure 1? Is threshold values of weight are region-specific? Which dataset is used to calculate the soaring index (S), convection index (Ko), total totals (TT)? What is the temporal and spatial resolution of those data?

We found that the variability of the Doppler velocity is the most reasonable measure to distinct between stratiform and convective rain. Therefore this measure was assigned the highest weight. The model data from COSMO (temperature and humidity profile) differ too much if the nearest model grid point is a grid point with precipitation or without precipitation. Therefore, the model derived measures as TT, KO, and So got smaller weights, because the measured variables are more trustworthy than the modeled ones. We added missing model information.

P4: Ln 21: On what basis the convection score partition (stratiform less than 3, inconclusive 3-6 and convective >6) is taken? Does the author consider the rain rate criteria also?

These values are chosen to have a big transition (inconclusive) zone between stratiform and convective. This is a very strict separation. In this way the probability of false classification is very low and the data set is prepared for training the algorithms. The huge amount of inconclusive cases is not part of interest here. We added missing information. Additionally, the upper threshold was set to 5.5 to increase the amount of profiles that are assigned as convective.

P5: Figure 2: The inconclusive data points are more than the convective and stratiform samples. Please comment on it? Whether the inconclusive samples are the transition from convective to stratiform event.

Yes inconclusive cases are the transition between both classes. See comment above.

P5: Ln 3: … visual classification of each single profile. How the authors have visually classify the profile into convective and stratiform? What parameter and criteria are used for the visual classification? Please provide a skill score table for better representation of your results.

We had a detailed view on randomly selected cases, if the classification using the convection score worked well. We checked the synoptic situation and had a look at the mentioned variables. We added missing information according to your suggestions.

P10: Figure 6: Out of two proposed algorithms (PDF and ANN), which method is superior? Authors also need to discuss the source of errors for each method. In the manuscript, the evaluation of the precipitation classification algorithms is not shown.

We restructured the algorithm comparison part with a new figure (Fig. 7, now Fig. 8). Thereby we improved the discussion of the results including misclassifications according to you suggestions.

Please provide some discussion on the proposed algorithm and related future research to put the results into a broader context.

We improved the discussion and outlook part.

**Minor:**

P2: Ln25: … following section. Change to sub-section.

Done as suggested.

P5: Ln 10: Zmax up to 50 dBZ. Don't you think there will be attenuation at such high reflectivity value?

We use the radar reflectivity factor $Z$ calculated from the drop size distribution (and Doppler spectra) and not the measured reflectivity $Z_e$. It is not affected by attenuation. Anyways, Zmax of up to 40 dBZ occured only rarely and do not influence the results significantly.

P6: It will be good to show the rainfall distribution like figure 3d.

We included Fig. 3 in Sec. 3.2 which shows the frequency distribution of stratiform and convective precipitation at 300 m height according to your suggestions.

P7: What bin size the authors have considered for Eq. (4) and (5).

We added the missing information to the text.

P11: Figure 7: PDF is overestimating the convective and stratiform precipitation than ANN. Which result is more accurate. For the inconclusive sample, both the methods have the same occurrence frequency. Why the number of data sample (NPDF and NANN) for analysis are different.

We restructured the comparison part and replaced Fig. 7 (now Fig. 8). We improved the discussion according to the reviewers suggestions.

**Anonymous Referee #2**

**General comments:**

The study discusses the two algorithms PDF and ANN for classifying convective and stratiform precipitation profiles based on MRR data. The authors utilizes the maximum reflectivity, mean Doppler velocity and maximum deviation in velocity within +/- 15 min. But there have been a numerous studies on this topic using various ANN based algorithms (e.g., Ghada et al., 2019, doi:10.3390/atmos10050251, Jergensen et al., 2020, DOI: 10.1175/WAF-D-19-0170.1).

We expanded the literature review.

The paper is topic of interest. The authors presented only two algorithms. It could have been good to show the results from various ANN based models to discriminate the convective and stratiform profiles and compare them.

This is not the scope of the paper. We aim for an algorithms based only on MRR observations to enable a wide spread and straightforward usage for ground-based remote-sensing sites.

Further, authors should include the validation metrics such as RMSE, MAPE, etc in tabular form for both the models.

The pdf method doesn't produce validation metrics. The results are visualized in Fig. 7 and discussed in the corresponding section. Nevertheless, we improved the discussion part by adding more information on the performance of both methods compared to the convection score. See more detailed comments below.

Perhaps, use of convolution neural networks (CNNs), Long-short Term Memory (LSTM) and recurrence neural networks (RNNs) will provide better forecast for time series data.

Yes, this could be. However, our methods provide satisfactory results.

However, I concern about following comments. I recommend that this manuscript requires major revision before its acceptance.

**Detailed comments are provided below:**

P2: Why authors are used two year data for training? Is this data covers the all dynamic ranges observed convection/stratiform? Any ANN based model, the training data should have the all range of values.

Actually, we use just one year 2013 for training. In principle you are right, more training data would improve the ANN accuracy. In future, we will apply the retrievals at our remote sensing site at University of Leipzig. Currently, we are observing 24/7 with our MRR. So the amount of data is increasing and we will have the opportunity to improve the algorithms with new data. We added that part about our future activities in the outlook of the paper.

P3: Are three indices such soaring index (S), convection index (Ko), total totals (TT) derived using COSMO model data? If so, is COMSO derived indices are validated with indices calculated form radiosonde observations?

Yes, based on COSMO-EU, but it is not validated with radiosondes since there are no radiosonde launches at JOYCE on regular basis. We added some information about the COSMO-EU model such as resolution.

P3:L5: Why the authors are used 15 minutes interval, where MRR gives 1 minute data?

We didn't use a 15 min time interval. We use 1 minute but calculate the temporal standard deviation of the maximum Doppler velocity (per profile) based on plus minus 15 minutes time span. It is a reasonable time span for convective rain.

P4.L6-7: … convective precipitation contains larger rain drops … Is it true always?
Include reference.

We added Niu et al. (2020) as reference and rephrased the sentence.

P4.L8-9: … +/-15 min is a reasonable time span for classification of rain events … But, there are the occasions, where the life time of convection will be less than 15 minutes. Authors should modify the sentence. Include reference.

In case of shorter rain showers the standard deviation is calculated on the shorter time span. In any case the standard deviation of the Doppler velocity is higher than during homogeneous stratiform rain. We added an explanation to the text.

P5.L9: … PDF and ANN method are based on training, the data has to be free of extreme or unphysical values … Do authors mean that the data cleansing? I understood that the data filter was performed in MRR data. If so, rewrite the sentence. However, what are the extreme values? Because, in general, if the trained data consists of all dynamics range, then the model will be able to predicted with better accuracy.

We rephrased the sentence to clarify misunderstandings.

P5:L24: What are the modification are done in Liu et al. (2004) and Liu et al. (2009) algorithms.

We rephrased the paragraph and added missing information.

P6:L23-24: … the confidence of a discrimination algorithm can be improved by using three measurement variables instead of only one or two … Are the proposed number of variables are sufficient? Is the model predicting better accuracy, if you consider more than three input variables? Is the prediction depends on number of depended variables? Why authors are not consider rain rate for training MLP model?

Yes, the model accuracy depends on the number of dependent variables. We do not consider rain rate, since rain rate and reflectivity are calculated using rain drop number concentration. There will be no additional independent information. We added these information to the text.

P8:L16: Is the MLP model is multivariate multi-step? I also suggest including a table with hyperparameters of MLP used in this study, rather than mentioned in the text.

Since the MLP is a categorization model the most meaningful quantity for the estimation of uncertainty are categorical cross entropy to calculate the loss, and the categorical accuracy. RMSE or MAPE are not suitable for the description of the categorical model since it gives probabilities to be stratiform, inconclusive or convective.

The authors did not give an overview about the hyper parameters in a table since Fig. 5 gives most information about the ANN setup. All other information such as accuracy are given in the brief ANN section.

P9: For readers, change line colour to red in figure 6 (m) & (n).

We improved the entire figure according to your instructions.

P9: I would rather suggest to include about "how often convective/ stratiform/inconclusive profiles occur at JOYCE supersite?

The numbers are given in Fig.2. We added that information for the test data set 2014 in the new figure 7.

P11: The relative occurrences of inconclusive profiles are equally weighting with convective/ stratiform in this study? Are they meant for transition profiles?

We completely changed and improved the evaluation of the algorithm performances. It is more clear now.

[revised manuscript text omitted]

---

## Referee Report (RR1)

AMT-2020-290

**Evaluation of micro rain radar-based precipitation classification algorithms to discriminate between stratiform and convective precipitation**

By Andreas Foth, Janek Zimmer, Felix Lauermann, and Heike Kalesse

==================================================================================

General comments:

This paper proposes to discriminate MRR measured rain patterns into stratiform, convective and inconclusive categories, based on the sole MRR observations. Therefore, it compares two dedicated approaches, i.e. Probability Density Functions or Artificial Neural Networks, to convection scores obtain using both MRR and COSMO model variables and defined and the reference classification.

The paper is of good interest, well written and easy to read. Overall, it carries the required information to understand the arguments developed but, although the methodology approaches are well described, sometimes the paper lacks necessary details and/or preciseness, especially concerning the rationale behind some choices.

Also, it appears that the article has already been through a review process and that the authors have provided a complete and significant response to the expert's comments during this review.

Specific comments:

Convection score section 3.2:
- one could precise that the weighting factors were define to obtain a total score comprised between 0 and 10.
- more importantly, one must argument the choices made for the respective weight of those factors (ranging from 1 to 3) as well as the threshold values (3 and 5.5) chosen to make the classification.
- some discussion or references regarding the defined transition values for each variable would be useful (for example on what base is rain stratiform for $\sigma v_{D,max}$ below 0.8 and convective above 1.2 ?)
- P5, L 18: give an estimate of the number (or relative number in %) of inspected cases for verification of the classification effectiveness

ANN section 3.4:
- One could have expected some further discussion about the network architecture was it predefined through *Tensor-Flow* or did you have options of number of layers and number of nodes, and if so, how did define the network used.
- P 10, L 7 and 12: reading those two passages set some confusion on the actual node output in the process: a value between 0 and 1, or values of -1, 0, or 1. May be the text could make the understanding easier

Conclusions

- P 13, L 4: reminding here the arguments leading to this assertion (i.e. "outperforms") would be useful to confirm this point
* * *
Conclusions:
* * *
This paper introduces a comparison of two interesting approaches for self-standing MRR classification of rain between convective and stratiform. It is well written and easy to follow hence it should make a nice contribution and find good use in the community.
Nonetheless, I believe that some arguments would benefit from more extensive justification and/or description of the rationale behind some set choices.

Thus, it is my recommendation that the paper be published after accounting for the requested minor revisions outlined.

---

## Author Response (AR2)

AMT-2020-290

**Evaluation of micro rain radar-based precipitation classification algorithms to discriminate between stratiform and convective precipitation**

By Andreas Foth, Janek Zimmer, Felix Lauermann, and Heike Kalesse

=====================================================================

**Response to referee #2 report**

We like to thank the reviewers for providing helpful comments to improve the manuscript.

We made minor changes according to the referees suggestions. All changes are highlighted in the diff-mansucript below. Added text is wavy-underlined and blue, discarded text is struck out and red.

The reviewer comments are listed below in black. The authors response is written in blue.

**General comments:**

This paper proposes to discriminate MRR measured rain patterns into stratiform, convective and inconclusive categories, based on the sole MRR observations. Therefore, it compares two dedicated approaches, i.e. Probability Density Functions or Artificial Neural Networks, to convection scores obtain using both MRR and COSMO model variables and defined and the reference classification.

The paper is of good interest, well written and easy to read. Overall, it carries the required information to understand the arguments developed but, although the methodology approaches are well described, sometimes the paper lacks necessary details and/or preciseness, especially concerning the rationale behind some choices.

Also, it appears that the article has already been through a review process and that the authors have provided a complete and significant response to the expert's comments during this review.

**Specific comments:**

Convection score section 3.2:

- one could precise that the weighting factors were define to obtain a total score comprised between 0 and 10.
  We added the value range information as suggested.

- more importantly, one must argument the choices made for the respective weight of those factors (ranging from 1 to 3) as well as the threshold values (3 and 5.5) chosen to make the classification.
  We added an improved argumentation for the choice of the weightings. The choice of the threshold values of 3 and 5.5 is already elucidated in an entire paragraph.

- some discussion or references regarding the defined transition values for each variable would be useful (for example on what base is rain stratiform for $\sigma v_{D,max}$ below 0.8 and convective above 1.2 ?)

  These values were determined empirically. We added a description to the text as suggested.

- P5, L 18: give an estimate of the number (or relative number in %) of inspected cases for verification of the classification effectiveness

  The relative number amounts to about 10%. We added the information as suggested.

ANN section 3.4:

- One could have expected some further discussion about the network architecture was it predefined through *Tensor-Flow* or did you have options of number of layers and number of nodes, and if so, how did define the network used.

  We added further explanations as suggested.

- P 10, L 7 and 12: reading those two passages set some confusion on the actual node output in the process: a value between 0 and 1, or values of -1, 0, or 1. May be the text could make the understanding easier

  We added two new sentences to clarify the confusion.

Conclusions

- P 13, L 4: reminding here the arguments leading to this assertion (i.e. "outperforms") would be useful to confirm this point

  We rephrased this sentence.

**Conclusions:**

This paper introduces a comparison of two interesting approaches for self-standing MRR classification of rain between convective and stratiform. It is well written and easy to follow hence it should make a nice contribution and find good use in the community. Nonetheless, I believe that some arguments would benefit from more extensive justification and/or description of the rationale behind some set choices.

Thus, it is my recommendation that the paper be published after accounting for the requested minor revisions outlined.